# "You're so powerless": Takatāpui/LGBTIQ+ people's experiences before becoming homeless in Aotearoa New Zealand

**Brodie Fraser**[ID]⊕*, **Elinor Chisholm**⊕, **Nevil Pierse**⊕

He Kāinga Oranga, Department of Public Health, University of Otago, Wellington, Aoteaora New Zealand

⊕ These authors contributed equally to this work.
* brodie.fraser@otago.ac.nz

## Abstract

Little is known in Aotearoa New Zealand about experiences of homelessness amongst Takatāpui/LGBTIQ+ identifying people, despite growing international literature regarding LGBTIQ+ homelessness. Using data from semi-structured interviews with eight people who identified as Takatāpui/LGBTIQ+ and had experienced homelessness, this paper explores their experiences prior to becoming homeless. These experiences are placed into the categories of: the pervasiveness of instability (especially in regards to family relationships, finances, and housing), having to grow up fast due to social and material conditions, experiences of looking for housing in stressed markets, and systems failures that resulted in a lack of autonomy. These results show that instability and systems failures are key contributors to Takatāpui/LGBTIQ+ people becoming homeless in Aotearoa New Zealand.

## Introduction

Experiences prior to becoming homelessness amongst LGBTIQ+ identifying people remain an emerging theme of homelessness research. This emerging research, in particular, focuses primarily on young people [1, 2]. The existing literature is thus in need of contributions which includes the experiences of both young people *and* adults. This paper fills this gap, focusing on an exploration of Takatāpui/LGBTIQ+ people's experiences prior to becoming homeless. Furthermore, this paper fills a second knowledge gap, namely that there is extremely limited Aotearoa New Zealand (henceforth referred to as Aotearoa) specific research published on experiences of homelessness amongst Takatāpui/LGBTIQ+ identifying people. Due to this limited existing research, this paper aims to explore the commonality of experiences prior to becoming homeless amongst Takatāpui/LGBTIQ+ people in Aotearoa; data collected for the wider project this paper comes from is sufficiently rich for us to focus this paper solely on experiences prior to becoming homeless. This paper will outline these common experiences, and then situate them in the existing international literature. It is worth noting that the aims of this research were: to explore the experiences of homelessness for Takatāpui/LGBTIQ+ people in Aotearoa; to investigate how Takatāpui/LGBTIQ+ homelessness relates to other sites of

**Data Availability Statement:** Data (i.e. interview transcripts) cannot be shared due to confidentiality. The Takatāpui/LGBTIQ+ community in Aotearoa New Zealand is so small that anonymity could not

be guaranteed even when identifying information is removed from transcripts. Thus we are unable to make data available in order to protect our participants. Ethics requirements for this study stated that only the research team (i.e. the authors) would have access to the transcripts. The University of Otago Human Ethics Committee gave ethics approval on this basis. For any questions contact gary.witte@otago.ac.nz.

**Funding:** BF, HC, NP, and EC were all funded by a New Zealand Ministry of Business, Innovation, and Employment Endeavour Grant. https://www.mbie.govt.nz/science-and-technology/science-and-innovation/funding-information-and-opportunities/investment-funds/endeavour-fund/success-stories/past-rounds/2016-successful-proposals/ The Funder had no role in study design, data collection and analysis, decision to publish, or preparation of the manuscript.

**Competing interests:** The authors have declared that no competing interests exist.

oppression; and to understand how both government and wider support systems shape the experience of Takatāpui/LGBTIQ+ homelessness. Furthermore, we further our understanding of the issue through the use of intersectional thinking. Intersectionality suggests that people experience different and multiple oppressions in response to their different identities which result in complex outcomes [3, 4]. Intersectionality encourages us to consider how upstream social determinants (such as racism, sexism, classism, transphobia, and queerphobia) form interlocking systems of oppression which shape the experience of people with multi-dimensional identities [5].

The acronym LGBTIQ+ stands for lesbian, gay, bisexual, transgender, intersex, queer, and other minority gender and sexual orientation identities (such as pansexual, non-binary, and asexual). The inclusion of the + sign is intended to be inclusive of the additional identities that are not included within the LGBTIQ acronym. Since the 1870s, the word takatāpui has been translated to mean "intimate partner of the same sex," and in the 1980s it was gifted to Māori LGBTIQ+ identifying people by academic and activist Ngahuia Te Awekotuku [6]. The word is now widely used among LGBTIQ+ identifying Māori as both an identity in and of itself, and as an umbrella term [6].

There is limited existing literature on LGBTIQ+ homelessness that is specific to Aotearoa. However, international literature finds that 20–40% of homeless populations are LGBTIQ+, despite this group only comprising an estimated 5–10% of the wider population [7]. Experiences of homelessness and LGBTIQ+ identity intersect to produce disproportionately high rates of issues such as: substance use, poor mental health, sexual abuse, foster care, family relationship breakdown, survival sex and sex work, and physical ill-health amongst LGBTIQ+ people who experience homelessness [1, 7–12]. Other key themes in the existing literature include: poverty, racism and ethnic discrimination, discrimination and stigma, and shelter inaccessibility [7, 13–17]. This international literature highlights how LGBTIQ+ identifying people are over-represented in homelessness statistics, and the range of negative outcomes they face.

In the Aotearoa context, rates of homelessness can be measured using data from the New Zealand Census. At present, the Census does not measure the number of transgender, non-binary, and gender non-conforming people in Aotearoa New Zealand. There are plans to change the way gender is measured at the next (2023) Census. In Aotearoa, homelessness is defined as "living situations where people with no other options to acquire safe and secure housing: are without shelter, in temporary accommodation, sharing accommodation with a household or living in uninhabitable housing" [18]. At the 2018 Census, there were over 41,000 New Zealanders, or nearly 0.9% of the population, who were homeless [19]. Of these, Māori (Māori are the indigenous people of Aotearoa) and Pacific people's rates of homelessness were nearly four and six times higher, respectively, than Pākehā (Pākehā is the te reo Māori/Māori language term for white/European New Zealanders) [19]. The 2018 Census data also showed that slightly more women were homeless than men [19]. The rates of homelessness in Aotearoa have steadily been increasing since 2001, when counts first began [19]. For Takatāpui/LGBTIQ+ people in Aotearoa, homelessness statistics do not yet exist. However, academic research on other aspects of Takatāpui/LGBTIQ+ people's lives is slowly increasing. Specific literature that looks at the health and wellbeing of Takatāpui/LGBTIQ+ communities explores themes such as youth suicide [20], psychological distress [21–24], drug and alcohol use [25], HIV prevalence and risk factors [26, 27], and non-consensual sex among Māori men [28, 29]. There is a general trend of Takatāpui/LGBTIQ+ people facing poorer health and wellbeing outcomes than their non-Takatāpui/LGBTIQ+ counterparts.

## Methods

### Sampling and procedures

Interviews with Takatāpui/LGBTIQ+ identifying people who had been, or were, homeless were conducted between October 2018 and February 2019. Participant recruitment was carried out via posters in key locations across Wellington, social media and emails, word of mouth, and researcher visits to additional key locations such as The Free Store (The Free Store redistributes unsold food from local cafes to the community, free of charge. People congregate outdoors to receive the food, and anyone is welcome to receive food) and Tīwhanawhana (Tīwhanawhana is a kapa haka and waiata group for LGBTIQ+ people; members are mostly Māori but non-Māori are also welcomed. It is known to be an important community space for the Takatāpui/LGBTIQ+ community) to talk to potential interviewees who may have experienced homelessness. The most successful form of recruitment was advertising on social media. Posters and word of mouth were also successful; researcher visits to key locations did not eventuate in any recruitments. Some potential key locations—such as soup kitchens and night shelters—were not visited. This was a conscious decision to seek out the viewpoints and experiences of people who might not have utilised housing service providers. BF was hesitant to align themself with service providers as they knew, from experience volunteering within the Takatāpui/LGBTIQ+ community for a number of years, that there is limited trust of organisations meant to provide support. We wanted to ensure we did not conduct interviews with any biases about how service providers were performing, and the support they were providing to participants. These values that BF held certainly influenced this selection criteria. However, this research utilised the critical paradigm, which allows for the researcher's values to be "central to the task, purpose, and methods of the research" [30]. This gives space for BF's insider position as a member of the Takatāpui/LGBTIQ+ community, and the values they attach to this, and enables them to understand the research findings on a level that an outsider to the Takatāpui/LGBTIQ+ community might not.

Eight people were interviewed who met the Aotearoa definition of homelessness [31]. Other inclusion criteria was that participants identified as being part of the Takatāpui/ LGBTIQ+ community, and located in the region in which BF lived (this was so that participants would be easily able to take part in a participatory video project that was to complement this research. However, as will be discussed in future publications, this additional project was not successful). Two participants were interviewed twice after we read the transcripts of the first interviews with them and realised there were important areas where the primary researcher could probe them for more information. This was in part due to BF's newness as a qualitative researcher, but also due to their insider position as a member of the LGBTIQ+ community, which meant that they presumed to understand what was being said, instead of probing for more detail [32]. Follow up interviews included asking for further details on participation in sex work, chronological timelines, and family relationships. One additional interview was conducted but this was not transcribed or used in the analysis as, over the course of the interview, it became apparent that the interviewee did not fit the Aotearoa definition of homelessness. There were three scheduled interviews that did not take place due to the prospective interviewees experiencing stress and personal crises at the times we had scheduled. Despite BF's best efforts, these were unable to be rescheduled. This highlights the precarious nature of the lives of many Takatāpui/LGBTIQ+ people who have experienced homelessness.

Ethics approval was granted by the University of Otago's Human Ethics Committee, reference number 18/147. Informed consent was obtained in written format. Each participant was gifted a $20 voucher of their choosing, and had a hot beverage and snack bought for them at the time of the interview if they wished. The interview schedule was piloted twice; however, it

was not followed in exact order as these were semi-structured interviews, meaning the schedule was instead a guide for BF as a new qualitative researcher to ensure they were covering all the topics they felt they wanted to ask about, and to provide them with prompts of how they might ask those questions. As it turned out, BF did not need to ask a lot of the questions on the schedule, as the participants had answered them in the course of the interviews without needing to be prompted. Each interview lasted about and hour, and were recorded and transcribed by BF, who had professional transcribing experience prior to beginning this research. The first 15 minutes of one interview were not recorded; the participant gave a brief overview of what he had said at the end of our interview and BF recorded the missed information in a memo directly after the interview.

## Data analysis

This research used constructivist grounded theory, as put forth by Kathy Charmaz in her book *Constructing Grounded Theory* [33]. Differing from traditional grounded theory, constructivist grounded theory acknowledges multiple standpoints, roles, and realities on part of both the research and research participants [34]. Furthermore, it adopts a reflexive stance towards the researcher's background, values, actions, and relationships with their participants, and situates their research in the historical, social, and situational conditions in which it was produced [34]. The use of constructivist grounded theory meant there was no target sample size set in advance, as this is determined by theoretical sufficiency [33, 35]. In grounded theory studies, sampling is 'driven' by the researcher's developing theory, and as such there is no prescribed number of participants at the beginning of the study [36]. Instead of providing an expected number of participants, researchers theoretically select participants who will be able to help construct the theory [35]. Data collection in grounded theory studies, then, ends when gathering new data no longer sparks new insights or reveals new elements of core theoretical categories [35]. Theoretical sufficiency is said to be reached when the data no longer produces theoretical insights. Once theoretical sufficiency was reached at seven interviews, the research team was consulted to confirm this was appropriate. An additional participant was interviewed to ensure theoretical sufficiency was reached.

Birks and Mills [37] outline the key components of grounded theory methodologies as the following: initial coding and categorisation of data, concurrent data generation or collection and analysis, writing memos, theoretical sampling, constant comparative analysis, theoretical sensitivity, intermediate coding, identifying a core category, and advanced coding and theoretical integration. A total of 126 codes were created. Of these, 72 were top-level codes and a further 54 were secondary-level codes nested underneath various top-level codes. Codes were created primarily by BF, and the remaining authors were consulted throughout. During the beginning of coding, the codes were mostly descriptors that frequently related to the findings of the literature review, and the interview schedule. As more interviews were conducted, categories were determined, and the data was coded accordingly. These intermediary codes/categories were discussed, refined, and expanded upon with the remaining authors. Following this, the data was coded again with a consciousness of the emerging theory. The emergent grounded theory will be presented and discussed in depth in future publications.

## Results

The results relating specifically to participants' experiences before becoming homeless are presented below. Due to the richness of data and lack of existing information about Takatāpui/LGBTIQ+ people's experiences of homelessness in Aotearoa, we choose to focus this paper on experiences prior to becoming homeless.

**Table 1. Demographics.**

| Pseudonym | Gender | LGBTIQ+ | Ethnicity | Age at Interview | Forms of Homelessness | Decade of Experience | Class |
|---|---|---|---|---|---|---|---|
| Avery | Gender fluid, Bigender, Female | Gender fluid, Bigender, Trans, Bisexual | Pākehā | 50 | Garage, Couch surfing, AirBnB | 2010s | Middle |
| Ayeisha | Female | Lesbian | Pākehā | 77 | Garage | 1980s | Working |
| Clara | Female | Trans | Māori | 37 | Emergency accommodation, Rough sleeping, Hostels, Couch surfing | 1990s, 2010s | Working |
| Felix | Male | Pansexual | Pākehā | 36 | Rough sleeping, Couch surfing | 1990s, 2000s, 2010s | Working |
| Marielle | Female | Queer, Pansexual | Pākehā | 25 | Rough sleeping, Couch surfing | 2010s | Middle |
| Nico | Takatāpui | Queer, Trans, Takatāpui | Māori | 35 | Couch surfing, Squatting, Bus/van, Foster care | 1990s, 2010s | Working |
| Omar | Male | Bisexual | African | 39 | Rough sleeping | 2010s | Upper-middle |
| Thom | Male | Bisexual | Pākehā | 42 | Rough sleeping, Couch surfing, Hostels | 1990s | Middle |

Table 1 provides basic demographic details for participants. As shown, there was a mix of ethnicities and ages; however, most participants were Pākehā (we have kept Omar's exact nationality vague in order to maintain anonymity), and in their late 30s. There was a mix of class backgrounds with several participants from middle class backgrounds, and one from an upper-middle class background. For the most part, participants were highly educated. Avery, Ayeisha, Marielle, and Omar all mentioned having attended university. Felix and Clara had attended polytechnics (Also known as Institutes of Technology, which are tertiary institutions in Aotearoa that generally offer hands-on, vocational, study options). Despite these varied backgrounds, all participants experienced severe poverty and financial insecurity before, during, and after their periods of homelessness.

As a note, social class is used in this paper to understand the social and economic backgrounds of participants. Class was determined by participants' parents' occupation and participants' level of education. Class was used as Socio-Economic Status (SES) is not a useful way to talk about people who are experiencing homelessness. SES measures the amount of money a person has and as all of the participants had experienced homelessness, they all had a low SES. We needed a better way to understand their economic and educational positions. We determined class by looking at the participants' education levels and their parent's jobs and education levels where we knew them. Our demographic does not reflect what we presume homeless populations to look like. For example, Avery and Omar both were private school educated and had relatively well-off families, but still became homeless, one because of transphobia and the other because of drug addiction. Class is a useful way of understanding how our participants bounced back out of homelessness and what their aspirations were. For example, a lot of them had middle-class aspirations of things such as home ownership or white-collar employment. Additionally, we chose not to collect Iwi (Iwi are tribes/nations in te ao Māori/the Māori world) data due to confidentiality reasons; the small sample size and small size of the Takatāpui/LGBTIQ+ community in Aotearoa means that participants would have had their anonymity jeopardised as they would have been easily identifiable.

These results include the categories of the pervasiveness of instability, having to grow up fast due to social and material conditions, experiences of looking for housing in stressed markets, and systems failures that resulted in a lack of autonomy.

## The pervasiveness of instability

Pervasive instability—particularly of family, finances, and housing—was a thread woven through the interviews; some interviewees explicitly discussed experiences of instability, and others alluded to it during our interviews. Thus, it was generated from data and from participants' own constructions. Instability was experienced primarily in childhood; it disrupted important periods of growth and learning about norms, and community connectedness provided a form of stability. Participants spent much time searching for, or trying to create, stability. Many of the interviewees saw instability as inherently negative, and stability as inherently positive. For them, instability—be it economic, housing, family, or employment related—was disruptive, and positioned them in a state of precarity. They found it impinged on their clarity of thought and encouraged impulsive, or even destructive, choices. Once in an unstable state, they frequently found it difficult to regain stability; instability was pervasive. Additionally, instability can be both sustained and episodic. The impact of instability is lasting and has multiple consequences that ripple throughout a person's life. This aligns with existing scholarship which shows that these forms of instability (economic, housing, family, and employment) have long-lasting impacts [38–43].

Family instability—such as under- or un-employment, changes in parental relationships, residential mobility, family violence, and parental addiction and mental health issues—can drastically upset children and adolescent's lives [41, 44–47]. Instability in childhood can then become the norm. Clara, Nico, and Felix all experienced instability at a young age. Not having models of stability in childhood and/or teen years made it difficult for them to know how to achieve stability, despite their innate understanding of it being a state to strive towards. Describing their teen years, Nico said:

> "And I mean everything. . .never really lasted long and. . .I think it was just some difficulties with stability. . .when you're not used to. . .living a stable life at all it's quite hard to make that for yourself."

Nico said their chronically unstable upbringing meant their parents did not model maintaining stability, particularly in relation to employment and financial stability. They explained this by noting "my parents didn't have normal employment. . .they were both self-employed, or unemployed." This economic instability had a ripple effect and they noted it became difficult for them to learn how to maintain stability within exploitative capitalist structures that facilitate precarity and intergenerational poverty. Instability in childhood can thus be difficult to mitigate.

Clara grew up in a rural town, and her father left her family when she was eight years old, which she thought was due to him not being supportive of the fact she is trans—she blamed herself for his departure. She found it hard to fit in and faced bullying at school. Family instability and disruption to her home life, and lack of a supportive community, contributed to her running away to Auckland at 14, plunging her into a period of chronic housing and economic instability. She said:

> "'95 [was] when I first went up to Auckland and then first got involved with sex work. . .for about four years. . .looking for places and couch surfing and staying in hotels. . .you can't have something [i.e. a tenancy] in your name when you're 14!"

Clara's childhood instability led her into an adult environment without knowledge of how to navigate it, which is directly linked to the theme of having to grow up fast. This instability in

her family life as a child had a compound effect and led to increased instability as a teen and adult.

Felix also experienced instability in childhood, caused by interwoven factors including an alcoholic father, his parent's strained relationship, and informal foster care from ages five to nine:

> "Not professionally, but. . .we kind of did week about. . .Basically so Mum and Dad could sort their shit, but then alternative weeks. . .Dad was there but in pickled state [drunk]. . . Mum would have the other kids there. . .they [the kids] just came and spent a week with us. . .and we got a break away from Mum and Dad and all their shit by going to their house."

In this case, alcoholism contributed to parental relationship breakdown, creating an unstable family life for Felix and his siblings; they frequently moved between different houses, and their parents were not able to consistently support and parent them due to alcoholism. Experiencing prolonged periods of instability during childhood and teen years meant instability became the norm for these three and has thus remained a disruptive factor into their experiences of adulthood.

During childhood, instability disrupted interviewee's growth and learning of norms; this resulted in them missing out on acquiring important knowledge or resulted in their expectations differing from the norm. Nico was one of the participants who missed out on learning important knowledge about things such as safety and tax codes. Nico ran away from home at age 14, and by 15, they were living in a van with their boyfriend and were having to teach themself new skills, such as cooking, without the guidance of adults. This resulted in them using a gas stove in an enclosed space:

> ". . .I had a boyfriend and. . .we lived in a van. . .I didn't know how you weren't supposed to cook things with gas in an enclosed space, now that I think about it, I'm like "how the fuck am I still alive?" You know, I s'pose when you're 15 you don't necessarily know about that sort of stuff."

Due to a lack of parental guidance, Nico had to grow up fast and thus did not have an adult around who was able to teach them basic skills. Instability during their youth also meant Nico did not have anyone who was able to teach them about tax codes upon entering the workforce, which resulted in them paying far more tax than they should have:

> "I didn't know about tax codes cuz I was a fucken child. . .my parents didn't have normal employment. . .so I didn't know about that sort of thing. . .I got that job and they were like "oh well if you don't have a tax code you have to pay 45% tax rate" and I was like "oh yeah, whatevs" so I was working like 40 hours a week and getting $150. . ."

The transition into adulthood is a difficult one in which young people are expected to learn new skills and gain practical knowledge in order to become capable adults. However, these examples highlight the ways in which instability can make it difficult for young people to learn these skills and knowledge sets.

Stability enables connection and growth, and stability in one area of life (such as housing) can make it easier to navigate challenges in other areas of life. Stability allows for support systems and community to flourish; without it one's sense of place and ability to plan for the future can be compromised [48]. Instability does not occur in isolation; disruption in one part

of life can lead to disruption in multiple parts of life. It is a complex, multi-faceted, experience that has the potential to disrupt people's lives on a significant scale. Part of its complexity is in how deeply connected it is to wider issues such as poverty, family relationships, stressed housing markets, and systemic failures. It is clear, then, why interviewees dedicated so much of their time, energy, and resources towards attaining stability.

## Having to grow up fast

Many participants shared experiences from their childhoods which forced them to grow up fast; these were frequently the direct result of the instability discussed above. Children and adolescents who experience undue hardship—such as poverty, family disruption, and extreme stress—often go through "subjective aging" or "subjective weathering" wherein they age faster than others who are the same age as them [49–51]. These terms refer to the "adultification" of children and adolescents whereby they are forced into adult roles and responsibilities at an earlier than expected age [50]. Hardships are seen to create older age identities because they "propel children and adolescents toward experiences that challenge cultural norms about childhood and adolescence" [49]. Furthermore, as seen above for young people who are homeless, early adultification can be a precursor to running away from home, and/or an outcome of running away from home [52]. Burton [53] outlines adultification as comprising social, contextual, and development processes whereby young people are prematurely and inappropriately exposed to adult roles and responsibilities. She outlines four typologies of adultification: precocious knowledge, mentored adultification, peerification/spousification, and parentification [53]. Many of these were visible throughout the data collected. Several interviewees discussed numerous experiences which indicated they were forced to grow up fast as a result of experiences during their childhood and teenage years. These include: living in fear as children, leaving home at a young age, and having sex at a young age.

Living in fear challenges the concept of childhood and as such, children and adolescents who experience non-normative levels of violence and feelings of being unsafe age more quickly in subjective terms [49]. For instance, Marielle faced abuse and family violence while growing up, the experiences of which led her to protect her younger brother from facing the same things:

> "I spent a lot of my childhood protecting him [her brother]. Like if he was gonna get a smack I'd jump in front to get the beating instead. Or if my parents were fighting really, really, aggressively I'd hide him somewhere and then go try and break up the fight. . .we don't talk about a lot of stuff. . .he's like "Oh nothing was that bad. . ." and I'm like "because you didn't see it, I hid you in a cupboard when things were going down, I hid you. . .so mum and dad couldn't find you, like you were being looked after" and so it's kind of hard for us to deal with the dichotomy of very different childhoods at the same time."

Marielle was forced to take on the role of protector of her younger brother in order to protect him from family violence; she subjected herself to unsafe situations so her brother would not face them. Living in fear and the adultification Marielle subsequently experienced meant she was not able to be a child. She and her brother viewed their respective childhoods differently, which put a strain on their relationship. This experience Marielle shared is an example of the parentification form of adultification, as outlined by Burton [53]. In this form of adultification, the child takes on the role of a parent to their siblings and/or parents [53]. We see in this example that Marielle serves as a protector and parent to her younger brother when her parents were unable to do so.

For Clara, adultification—specifically, feeling older than she was—came when she ran away from home at 14, due to the struggle of being a trans teenager in a small town with limited support and resources available. She said: "...I guess I would have to say I went from being a child to an adult...there was no teenage years for me." Adultification meant she felt as though she missed out on having a normal teen experience. This is consistent with literature that shows adultified young people often struggle to maintain recreational relationships with their peers [53]. In addition to this, her early adultification resulted in her being taken advantage of financially:

"I met up with some trans people...they'll take you under their wing as like your drag daughter, they'd be your drag mother...through them you would go out and work on the streets, earn money, and...your drag mum would take your money and use it for your accommodation and for whatever else they wanted...they would just like take all your money off you and give you hardly anything back, but not that you really knew that at the time...I was like 14...and didn't really click on to stuff like that."

This shows that despite Clara taking on adult roles, and being exposed to precocious knowledge, she was still a vulnerable teenager in a new, adult, environment. Adultification does not allow young people the luxury of finding independence at their own pace, instead pushing them into adult activity at a rapid speed, without them having an adult understanding of the events taking place.

Leaving home early also resulted in Felix having to grow up fast in order to be able to navigate the challenges he faced. The hardships he faced in childhood and adolescence resulted in his adultification and subjective age increasing. The main hardship, he noted, was being kicked out of home on his 16th birthday due to a homophobic stepfather:

"...my mother's partner, now her husband, didn't want a faggot, his words...I would have been kicked out of home the day after my 16th birthday, but the road was closed...the moment the roads re-opened, boom...[he kicked me] outta there, so yep, probably one of the longest days of my life."

This forced Felix to become an independent adult almost overnight; it propelled him into adult roles and responsibilities while he was a high school student. This is a clear example of subjective aging and the ways in which young Takatāpui/LGBTIQ+ people are forced into early adulthood in order to survive heteronormative and cisnormative environments. He found a job and began to work an increasing number of hours to provide for himself, which became too much to balance with school: "[I] moved to a flat and started a job at The Warehouse. The job hours went up and the schoolwork went down...I got through sixth form, so what's the point of going back?" (sixth form is what the second to last year of high school used to be called in Aotearoa. It is now called Year 12. Teenagers in this year are usually aged 16 or 17). Here, his immediate survival needs outweighed his educational needs; as a teenager he took on the responsibilities of an adult in order to provide for himself. In particular, living in poverty and having to be financially responsible for himself was his main catalyst when making decisions. These experiences influenced how he handles adversity: "with my childhood I felt very kind of blasé about a lot, it was just like life's gonna go this way, life's gonna go that way, just enjoy the ride and see where we end up." This resulted in him feeling as though he lacked autonomy; he noted that it took a long time to him to feel as though he had control about how his life went. Adultification and subjective aging caused Felix to experience non-normal teen years, as well as perceived lack of autonomy, well into adulthood.

Sex was another way in which some participants—particularly Nico and Clara—grew up fast; interviewees all had very different attitudes to sexual practice, and these two were two of the interviewees for whom sexual practice was linked to growing up fast. For Clara, this involved entering the sex work industry as a teen; for Nico, it involved the relationship between sex and power dynamics. Clara entered into sex work shortly after leaving home as it was the only way for her to get money without any qualifications at a time when she was unable to get any government support. When probed about what this was like, she said:

> "[It was] cool, cuz I had all this money. But also, also freaky, especially when you're like underage and suddenly you're introduced into doing very adult things like having sex with men and giving blowjobs. . .it's really very different when you're 14, but you just think about the money."

The survival strategies Clara undertook increased her subjective age. Having to learn adult practices such as sex was a "freaky" experience for Clara and it, along with her homelessness and poverty, forced her into early independence. To cope with this, she focused on how sex work was enabling her to collect an income at a time when she had no other support. This example shows one of the many forms subjective weathering can take; early sexual debut and homelessness are noted forms of subjective weathering and adultification [52, 54, 55].

Nico was forced, at an early age, to learn that sex could be used to exert power over people, and could be used as a form of power for themself:

> "I liked sex, but more than liking sex I liked saying no. . .I liked people wanting to have sex with me and me being like "nah.". . .I think that there was a lot of like power play in that. . .you don't have a lot of power to choose what happens in your life when you're really young because people don't listen to you. . ."

This quotation highlights both how difficult it was for Nico to navigate adult spaces as a teen who had experienced a significant amount of subjective aging, and how they compensated for that lack of agency and stability by finding power in saying no to sex. They enjoyed sex *and* enjoyed saying no to sex as they wanted more agency at this time in their life. However, they were not always able to say no when others tried to exert power over them via sex:

> "the foster mum (Nico had arranged informal foster care for themself in order to be able to receive financial support from the government) totally set me up to have sex with her brother who was like 25 or something and I was like "I am not into him.". . .[she] ended up getting me to go and hang out with him at his house in the middle of nowhere, it's not like I could go home at the end of the day because it's not like I had a fucking car, and oh there's only one bed there, obviously I'm supposed to have sex with him."

Here we can see that while Nico was cognizant of the power relations that can be intertwined with sex, but not always able to utilise them in a way that afforded them agency. They had been forced, through the circumstances of childhood instability and early homelessness, to enter adulthood earlier (i.e., experience subjective aging) and experienced the precocious knowledge form of adultification, particularly in relation to sex. Despite this, they were still expected to defer to older adults, as we can see in the above example, which limited their agency. Adultification/subjective aging and sex are thus closely connected to agency and power relations.

This section has examined the ways in which participants were forced to grow up fast, and the impact this had on their lives. Adversity in childhood forced participants to grow up fast, and resulted in them aging more quickly than their peers. Understanding experiences in childhood and teen years are necessary in being able to fully understand how and why Takatāpui/LGBTIQ+ homelessness manifests.

## Difficulty in finding housing

Participants shared experiences of looking for housing that took place in stressed housing markets, and included lowered expectations, discrimination, presenting as a different self for acceptance, and systemic issues. These experiences occurred primarily in adulthood, but were also present in adolescence for some participants. Finding housing caused considerable anxiety for participants and fed into existing instabilities, as well as diminishing their sense of self. Housing markets are a key factor that influence experiences of homelessness; highly competitive, expensive, and discriminatory housing markets can directly lead to episodes of homelessness [56]. Thus, marginalised people and low-income earners can find looking for housing to be incredibly difficult, and stressful, experiences. The experiences discussed below highlight how the housing system can fail those in need and contribute to homelessness.

Many interviewees were looking for housing in places where the housing market was already highly competitive and expensive—such as Wellington and Auckland. House prices in Aotearoa have steadily risen over the past 30 years, as have the number of households renting, with low-income households the most likely to be renters [57]. Affordable housing is lacking, and social housing (we use social housing here to refer to subsidised housing provided by the state, councils, and community organisations) as a proportion of total housing stock has steadily declined [57]. These factors are producing increasingly competitive housing markets. Such competition, particularly in the rental market, is likely to unduly impact on low-income earners as rents continue to increase. This makes it harder to obtain housing and plunges low-income earners into homelessness. Participants discussed their experiences with stressed housing markets. For instance, Avery described how, before she transitioned, she and her (now ex-)wife had struggled to find housing as well-paid public servants:

> "What really surprised me was if I go back to the early years of my marriage, say there's two of us, both. . .quite well paid, and we're looking for a rental place. . .even for us it was really hard, two relatively typical, respectable [people], paying the rent's not gonna be an issue. . .every property you looked at there was ten people there. . .[all] married couples with jobs. . .if it was that bad in that situation, it was gonna be almost impossible as a queer person looking for rental accommodation."

This quote acknowledges the stress that already existed within the Wellington housing market over a decade ago. Avery also provided a description of the ideal tenant she thought landlords were looking for (i.e., married, middle class, "respectable"), which she no longer conformed to as an openly—and visibly—queer person, with a low income. Such difficulties can directly lead to homelessness; such competitiveness in housing markets mean that those who are visible others—particularly visibly transgender people—are less likely to find housing, and thus at greater risk of becoming homeless. This is precisely what happened with Avery; she became homeless due to discrimination within housing and job markets.

Stressed housing markets resulted in participants lowering their expectations of what they would be allowed to get in the housing market, based on what they believed landlords were looking for. Participants' explanations of their difficulties finding housing revealed that they

drew on their life experiences, and understanding of the world, in order to make judgements about how they should navigate this. For example, Nico lowered their expectations of the rental market, explaining:

". . .I can write really well and speak in a way that everybody. . .feels like they wanna trust me. . .[I have a] stable job, like all that sort of stuff, I have a family, you know like I don't drink, I don't do drugs. . .I go to fucking bed early. . .technically a tenant like me is the sort of person everyone wants to rent their house to but because I'm trans people don't wanna rent their house to me."

This disjuncture between Nico's sense of self and the way they were treated while looking for housing resulted in them reducing their expectations during subsequent encounters with the housing market. Thus, Nico recognised that they were judged by landlords, and they came to realise this made it more difficult for them to secure housing. This difficulty resulted in Nico becoming homeless again in adulthood, this time with their child.

Clara also touched on this, noting that in her experience, landlords seemed dubious taking on trans people as tenants:

"I think landlords are really dubious about transgender people. . .I don't know what it is, I think it falls into that [too] hard basket category again where. . .they don't wanna bother, they don't wanna know about it, cuz it's not the normal."

Being trans, and thus a visible "other," meant Clara's experiences of finding housing have been characterised, in her opinion, by landlords making judgements about her being "too hard" to deal with, as she is not "normal." Through the lens of Goffman [58] we see that being a visible "other" here is a discredited stigma; meaning it cannot be easily hidden, and thus causes people to pass judgement on the stigmatised person. Feeling as though others are viewing her as difficult has resulted in Clara reducing her expectations both of landlords, and of the housing market more generally. Furthermore, such discrimination made obtaining housing a homeless transgender woman even more difficult than if she had been cisgender.

The subjective experience of participants as Takatāpui/LGBTIQ+ people looking for housing was that many of their interactions were characterised by them as discrimination. It is difficult to know if these interactions did involve overt discrimination, or if interviewees had perceived the situation this way due to the culmination of their past experiences, perceptions of others, and expectations of how they were going to be treated. For instance, Clara discussed how she felt she was discriminated against when she was evicted from a rental property:

". . .we'd end up getting a place. . .eventually we'd have about four or five of us staying in like a two or three bedroom house cuz there's. . .not really any other accommodation out there for trans people. . .We would eventually have people couch surfing at our places. . .that would often get us into trouble, like the landlord would say "oh I hear you've got heaps of people here, or youse are up really late at night time, what are youse doing?". . .And it's like well actually we're all sex workers so you know, we're getting the bills paid. . .a lot of the time they wouldn't like that. . .even after the law reform bill (the Prostiution Reform Act 2003, which decriminalised sex work in Aotearoa) went through it still wasn't really acceptable. . .then it was like "right, you've got a month to get out, youse have to leave. ""

Clara understood this situation as being discriminatory; she was kicked out of her rental homes because she and her flatmates were sex workers which she felt like her landlord

disapproved of, despite sex work making it possible for them to pay their rent. However, we do not know the perspectives of her flatmates, neighbours, and landlord—she noted she and her flatmates kept late hours, and had many guests, which could have made them undesirable as neighbours. While we are unsure of the exact year this example occured, it is highly likely it occured during a time when no-cause evictions were legal in Aotearoa, meaning it would have been legal for Clara's landlord to evict her if they had wanted to. These different norms around what is—and is not—acceptable contributed to friction in Clara's experiences as a tenant and neighbour. This rental instability caused Clara significant stress, and had the potential to result in her again experiencing homelessness.

Due to negative experiences while finding housing, and experiences of discrimination, there were instances where interviewees presented as a different self in order to meet their internalised beliefs about what they thought landlords/flatmates were looking for. When Marielle was looking for a flat, while homeless, she pretended to be straight during flat interviews:

> "you just kind of get very good at making yourself fit whatever mould they need to find. Like I moved into a flat where I knew they wanted a straight female, so I was just like yep, only like dick."

Her prior experiences of finding flats meant she was aware of the social norms around what makes one a successful flatmate candidate, and she adjusted how she presented herself accordingly. In addition to this, her desire for her episode of homelessness to end meant that hiding this part of her identity was a sacrifice she was willing to make. Similarly, Ayeisha noted when she was looking for housing "I knew. . .to be a little discreet about being a homosexual." This was due to her experience of being evicted from the garage she had been living in for what she suspected was homophobia: she said her landlords had suspected she was a lesbian and found a reason to end her tenancy. When she did eventually find housing, she made sure her new landlords did not know she was a lesbian. These examples show how Takatāpui/LGBTIQ+ people have had to hide their identities as best they can in order to navigate the housing market. These examples emphasise how some LGBTIQ+ identities are discreditable stigmas i.e.; they can be concealed and hidden. A further discussion of participants presenting as a different self for acceptance will be provided in future papers.

There was also an acknowledgement of systemic issues within Aotearoa housing markets; which directly links to the systems failures theme below. Specifically, Nico highlighted how those who are visible minorities *and* low-income earners are going to get locked out of competitive housing markets. They explained this was because, in their experience, landlords and property managers would be more likely to choose someone who is more like them (i.e., usually cisgender and Pākehā). They said:

> ". . .if you've got like $1500 a week to spend on a house you can have any house. . .but if you are competing with other people for cheap housing, they are always gonna choose someone who is like them, not someone who is like me [Māori and trans/takatāpui] and people like me are not usually landlords."

The intersectional combination of being a visible "other" and living in poverty was—and still is—systemically locking trans people out of the housing market, making them increasingly vulnerable to becoming homeless. Poverty is one of the main catalysts to homelessness and when it is combined with racism it can systemically lock people out of housing, pushing them into homelessness [13, 59–61]. This multiplicity of factors positions an already marginalised demographic in an unnecessarily precarious place within housing markets.

In sum, difficulty finding housing was a key category throughout the interviews. Navigating stressed housing markets was a difficult endeavour for many of the participants, which resulted in them lowering their expectations, experiencing stress and discrimination, and presenting as a different self for acceptance, all while trying to deal with systemic failures within the housing market. This was particularly evident amongst the transgender and gender diverse participants; their inability to fully conceal their transgender identity meant that landlords and flatmates were easily able to discriminate against them as visible "others."

## Systems failures resulting in a lack of autonomy

There were a number of systems failures that consequently restricted participants' autonomy; specifically, systems of housing, childhood/family, economic equality, and Takatāpui/LGBTIQ + inclusion and acceptance. These failures occurred both in childhood and adulthood. The failures of these systems served to further marginalise participants, and the instability, stress, and limited options they created served to restrict participants' autonomy; they do not have the ability to self-govern and the choices available to them are heavily restricted due to these systems failures. This category is linked to pervasive instability and difficulty finding housing. In exploring how different systems fail, and the ways in which they limit autonomy, we can further understand how Takatāpui/LGBTIQ+ people experience homelessness.

Aotearoa has a welfare state and social housing to provide support for people in need and provide a "safety net;" tenants are afforded protection through government legislation such as the Residential Tenancies Act; unemployed and low-income earners are able to receive a benefit and an Accommodation Supplement if their housing costs are high; and government funded social services provide support for issues such as housing, tenancy, and food insecurity. Additionally, charitable services are widespread, and often receive government funding. Despite this, participants observed widespread failings in all of these systems, which resulted in a lack of autonomy. Homelessness is a form of "structural disempowerment" which places people in a position of marginality [62]. Homelessness is a process and experience of disempowerment, which can serve to remove autonomy. As discussed earlier, there are numerous issues with the housing and rental markets in Aotearoa. One of these relates to the ways in which the system locks people out and reduces their autonomy. For instance, housing and rental systems failures have resulted in Avery having little choice about where she lives, and who she lives with, increasing the instability she faces. After an extended period of homelessness—which was triggered in part by discrimination in both housing and job markets, as well as lack of affordable housing stock—she finally found a flat to live in:

> ". . .now I'm back to flatting again, and you know, we've got a very well-known company as our property manager and they're very slack, and my flatmate a couple of times has kind of put her foot down and got some stuff done, but I'm afraid of rocking the boat, cuz I'm looking at the property from an economical point of view. . .somebody should buy it and redevelop it because it's a prime site. . .so I don't want to end up being like painful tenants to motivate people to sell the property from under us. Like, you're so powerless."

Going flatting was the only feasible housing option available to Avery, but, as shown, she found herself in a situation where she felt "powerless" to ask the property manager to address any of her concerns about the condition of the house; she was worried asking for repairs would jeopardise her tenancy and make her landlord consider selling the property. This is a common problem for renters [63]. This lack of power results in instability and transience amongst an already vulnerable population. Issues such as these within rental markets, and

minimal protections for renters, can lead to homelessness amongst populations who are living in poverty and facing intersecting oppressions such as those who are Takatāpui/LGBTIQ+ and Māori, if they attempt to challenge their landlords.

Systems failures of support structures during childhood also limited autonomy. For instance, Nico explained how the unstable nature of their childhood and lack of support and guidance during this time meant they struggled to know how to make good decisions, and had not learnt specific norms:

"I mean everything, it just never really lasted long. . .I think it was just some difficulties with stability. . .when you're. . .not used to living a stable life. . .it's quite hard to make that for yourself. . .like not having blueprints. . .a lot of times. . .I just didn't make good decisions. . ."

Here they are suggesting that not having "blueprints" and missing out on the stability and norms most experience during childhood has contributed to their poor decision making, which further entrenched the instability they experienced. The systemic failures that led to instability in Nico's childhood—such as entrenched poverty and racism—has had a significant impact on the trajectory of their life and made it difficult for them to make good decisions. Nico was able to recognise that had there been systems in place (such as better income support for their parents and themself and educational support to make their transition into adulthood easier), they might not have experienced such extreme instability.

Poverty and income inequities also resulted in a lack of autonomy for participants; specifically, the intersections between poverty, Takatāpui/LGBTIQ+ identities, and racial/ethnic minorities. Nico explained income inequities and discrimination that leave Māori trans people particularly vulnerable:

". . .a lot of trans [people]. . .are also sex workers. . .[and] developing your skills over a longer period of time doesn't mean you get more work, doesn't mean you get paid more. . .like for most people. . .by like the time that they're older usually they've been able to accumulate resources. . .for Māori trans people they usually haven't been able to accumulate that money. . .as they get older they get less money because they get less work and. . .maybe they are on a benefit but then that's hardly any money. . .they don't have a lot of options. . .there's very few older trans people who have managed to buy a house for themselves, and even if they could manage to. . .do lots of sex work and save ten grand or twenty grand they spend it on surgery (gender affirming surgery). . .on their basic healthcare needs. So, there's like a real systemic problem there."

The intersections of racism and transphobia, as well as poverty and the neoliberalisation of the welfare state, have resulted in Māori trans people being pushed into precarious living situations and cycles of poverty. This shows how intersection oppressions can result in unique, complex, problems for people, which Crenshaw [3] notes is important to embrace in order to resist compartmentalising experiences and attempts to undermine collective action. Furthermore, we can see that their autonomy becomes limited and, as Nico noted, they "don't have a lot of options." This, in turn, can lead to homelessness and make it difficult for an individual to exit homelessness and poverty without any additional support.

Systems failures that result in poverty and income inequities were also briefly discussed by Clara and Felix. Clara focused on how transgender people are judged for making choices in situations where their full autonomy is limited. She said:

"often transgender people are pushed up against a wall and then when we have to have make bad choices for ourselves cuz there's nothing else to do, then society looks down on us for making those choices when they've actually put us in those positions to begin with."

The transphobia she faced—and the lack of resources and support that accompanied it—meant she was not always able to make choices that she would have liked to have been able to make. The various systems failures Takatāpui/LGBTIQ+ people are facing mean, as Felix noted, "people are giving up their vocations, jobs, studies, just to have a safe roof over their head."

Following on from the above examples, insufficient support for Takatāpui/LGBTIQ+ identities was another key area in which participants experienced systems failures; some of the trans participants, in particular, faced a lack of support for their transition processes. People who are transgender face difficulties with discrimination and stigma, family rejection, and violence, amongst others [64, 65]. Transitioning is often a difficult process, particularly when there are minimal supports available to the individual who is transitioning. Transitioning combines social, medical, psychological, and interpersonal changes, and is a process which is often traumatic [66]. The ways in which participants' experienced Takatāpui/LGBTIQ+ specific systems failures were related to the lack of support available to transgender people when they transition. Clara struggled to have her needs met as a teenager when she first started to transition:

"I didn't really feel supported or have the available, available resources to like fully come out and be a transgender person or have your family be there to support you, like I felt completely isolated from everything."

This lack of support and failure to have her needs met were the catalyst in her becoming homeless; as mentioned earlier, Clara became homeless when she ran away from home at age 14. Clara now volunteers with a charity that supports people who are transgender—and other gender minorities—with any unmet needs they might have, such as accessing information about healthcare and changing identity documents. This example clearly shows how systems of social support and healthcare restricted Clara's autonomy; not having systems of support meant she was not able to be fully supported by her family and wider community when she came out. Ultimately this became intolerable to her, and she felt her only option was to run away from home.

Similarly, Felix mentioned how there was minimal support for him as a young person exploring his sexual orientation. Listing youth focused organisations around the country, he noted "none of these places were around when I was a kid." This absence of both family acceptance and peer groups meant there was no support for him when he first came out (and was consequently kicked out of home). This meant he was isolated from his community during a time when he was exploring his identity and trying to be his most authentic self. He went on to discuss how there are now a number of support and peer groups for Takatāpui/LGBTIQ+ teenagers and university students, but there is a lack of peer support groups outside of that. Felix decided to try fill this gap himself and re-start a peer support group for Takatāpui/LGBTIQ+ adults.

In sum, there were failures of multiple systems of housing, income, Takatāpui/LGBTIQ+ support, and family that served to restrict the autonomy and freedom of choice available to participants. These failures increased the instability interviewees experienced, and directly contributed to them becoming homeless. Survival within unequal structures restricts a marginalised person's agency through the limited range of options available to them.

## Discussion

The categories presented above give us a means through which we can understand specific experiences of homelessness and all it entails—such as poverty, poor mental health, drug addiction, and sexual practice. The following discussion increases our understanding of how homelessness was experienced and understood by participants; we can see how individual experiences feed into a larger system of inequities, missed intervention points, and inadequate support. These experiences prior to becoming homeless are consistent with international literature that shows LGBTIQ+ people experience structural (e.g., cisnormativity, heteronormativity, discrimination in the housing market, and having to grow up fast) and intrapersonal factors (such as addiction and poor mental health) which contribute to them experiencing homelessness [67–71].

Takatāpui/LGBTIQ+ people are frequently forced to hide their Takatāpui/LGBTIQ+ identity/ies simply to gain acceptance in housing and job markets, and in social service organisations such as homeless shelters [15, 16, 65, 72]. This goes against one's rights of freedom of expression, access to social security, and equal treatment. For example, the difficulty in finding housing section shows that overly expensive and competitive private housing markets forced many participants to present as a different self in order to try and obtain housing. This is consistent with international literature indicating that LGBTIQ+ people face discrimination within housing markets on the basis of their sexual orientation and/or gender identity [73–75]. These circumstances, and the survival strategies that they entailed, limited participants' freedom to present and act in ways that were consistent with their sense of self. Survival within unequal structures restricts a marginalised person's agency through the limited range of options available to them. Specifically, Takatāpui/LGBTIQ+ people experiencing homelessness are often forced to hide their Takatāpui/LGBTIQ+ identity/ies. Survival can often mean they do not have the choice to express their Takatāpui/LGBTIQ+ identities; the choice to live openly and proudly is not available to them without directly risking their safety, incomes, housing, and interpersonal relationships. Being forced to mask one's Takatāpui/LGBTIQ + identity places stress on the individual, which can carry through to their familial, social, and romantic relationships. Furthermore, this can intersect with other oppressions, such as racial discrimination, which can further limit autonomy [13]. For example, both racial and anti-Takatāpui/LGBTIQ+ discrimination in the housing market limits the range of housing options available to a person, and can result in insecurity and homelessness; this compounds for people with multiple vulnerabilities (e.g., those who are poor *and* Takatāpui/LGBTIQ+). Thus, people who face intersecting oppressions have limited agency by way of these structures of oppression forcing them to hide their most authentic self. Intersectionality thus helps us to see how multiple oppressions and the complexity of their interactions can result in negative outcomes [4].

Instability was a pervasive theme throughout each interviewee's narrative. This was primarily experienced in childhood, and had flow-on effects throughout their lives. Such findings align with international literature which indicates that people who identify as LGBTIQ+ and have experienced homelessness have also experienced a great deal of instability (particularly housing instability) throughout their lives [76–78]. Krause et al. [78] found that instability for LGBTIQ+ youth was rooted in early life experiences; this aligns with the findings presented in this paper showing numerous and wide-ranging instabilities experienced in childhood. Furthermore, the international literature shows that instability is worsened for those who hold multiple stigmatised identities, and that those who experience it face greater level of discrimination on the basis of LGBTIQ+ identity [76, 77]. These findings, along with the corresponding international literature, show that there are a number of common experiences amongst LGBTIQ+ people prior to them experiencing homelessness. This gives us important insights

into the multitude of factors involved in a person's experiences of homelessness; we can see the importance of structural factors such as residential and financial stability, safe and stable child-hoods, and well-functioning housing markets as protective factors against homelessness.

Another key finding was failures of institutional support systems. Takatāpui/LGBTIQ+ peo-ple experiencing homelessness were failed by institutions (such as government agencies and social service organisations) which contributed to, or worsened, their episodes of homeless-ness; social welfare was an area where participants were acutely failed. Part of this stems from the structure of the welfare system in Aotearoa. Rhetoric of welfare dependency and individual responsibility are a common and long-standing means through which governments can weaken welfare states. Narratives of welfare dependency link the security extended to individu-als to a loss of individual responsibility [79]. In Aotearoa, multiple governments and politicians have repeatedly justified efforts to re-shape the social welfare system as attempts to mitigate welfare "dependency" [80–84]. Despite this, government data indicates that benefit sanctions do not work, and only serve to increase the length of time spent on a benefit [85]. As this paper has shown, people who experience homelessness need, but do not currently receive, adequate income support to mitigate the poverty they experience. These findings show just how govern-ment and mismanaged social support systems have contributed to experiences of Takatāpui/LGBTIQ+ homelessness. We suggest that WINZ needs to have more flexibility built into its systems to take into account people's individual circumstances, particularly for young Takatā-pui/LGBTIQ+ people who have become homeless due to discrimination from within their families. The findings we present show real problems with the way Takatāpui/LGBTIQ+ expe-rience WINZ. This urgently needs to be addressed by government. Furthermore, we believe that implementing all of the recommendations made by the Welfare Expert Advisory Group [86] will ensure that individuals and their families are less likely to experience multiple systems failures, which, as we have seen, plunge people into a state of instability, entrenched poverty, and precarity; all of which can lead to homelessness.

Shelton [87] argues that those of us working to address Takatāpui/LGBTIQ+ homelessness must not problematise the individual, as this shifts our focus away from the root causes of the problem—such as heterosexism and queerphobia, cisgenderism and transbias, poverty, and racism—onto an identifiable villain. The main strength of this paper is that while it has given space for individual experiences to be explored, it has made clear the systemic failures and oppressions that lead to Takatāpui/LGBTIQ+ people becoming homeless. Shelton [87] notes that part of our work in preventing and ending homelessness is to help dismantle oppressive systems that privilege some citizens over others; this paper has contributed to this work. We have further contributed to this by considering the intersectional nature of participants' expe-riences. However, the main limitation of this paper is the homogeneity of participants. There were no disabled, intersex, Pasifika, or refugee participants, which means their experiences of Takatāpui/LGBTIQ+ homelessness in Aotearoa did not contribute to the research. Further research is needed to ensure these voices are heard. There are numerous additional avenues for future research about Takatāpui/LGBTIQ+ experiences of homelessness in Aotearoa. Some of the fundamental areas that need researching are; the frequency and demographics of Takatāpui/LGBTIQ+ homelessness, Takatāpui/LGBTIQ+ people's experiences with homeless-ness service providers, and Takatāpui/LGBTIQ+ people's routes into and out of homelessness. There is also a need for specific research on Māori Takatāpui/LGBTIQ+ homelessness.

## Conclusion

This paper has explored key experiences before Takatāpui/LGBTIQ+ identifying people in Aotearoa became homeless. Participants reported high levels of instability, particularly in

childhood. Alongside this, participants also experienced situations which forced them into early adultification, often due to the multiple systems failures they faced. Additionally, participants also reported a number of difficulties in navigating stressed housing markets, which often resulted in them having to present as a different self in order to secure housing. These findings look at a multitude of factors that contribute to Takatāpui/LGBTIQ+ people's experiences prior to becoming homeless, which broadens our understanding of the issues and contributes to existing knowledge about what systems changes need to occur in order to prevent LGBTIQ+ homelessness. This paper fills an important gap in the Aotearoa homelessness literature, and furthers international understandings of LGBTIQ+ homelessness.

## Acknowledgments

We acknowledge the research participants who generously shared with us their time and stories. It was a privilege to be trusted with their stories. Ngā mihi nui. We also wish to acknowledge transgender, non-binary, intersex, and gender expansive people's right to self-determination.

## Author Contributions

**Conceptualization:** Brodie Fraser, Elinor Chisholm, Nevil Pierse.

**Data curation:** Brodie Fraser.

**Formal analysis:** Brodie Fraser.

**Funding acquisition:** Nevil Pierse.

**Investigation:** Brodie Fraser.

**Methodology:** Brodie Fraser.

**Supervision:** Elinor Chisholm, Nevil Pierse.

**Writing – original draft:** Brodie Fraser.

**Writing – review & editing:** Brodie Fraser, Elinor Chisholm, Nevil Pierse.

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
