## [Decision Letter · Decision Letter 0]

17 Sep 2021

PONE-D-21-26555“You’re So Powerless”: Takatāpui/LGBTIQ+ People’s Experiences Before Becoming Homeless in Aotearoa New ZealandPLOS ONE

Tēnā koe Dr Fraser,Tēnā koutou Dr Fraser's co-authors,

Ngā mihi maioha; thank you for submitting your manuscript to PLOS ONE. After careful consideration, we feel that it has merit but does not fully meet PLOS ONE’s publication criteria as it currently stands. Therefore, we invite you to submit a revised version of the manuscript that addresses the points raised during the review process. These are elaborated on below, but the feedback is largely on expanding the methods and discussion, with no critical inclusions. The intention is to progress your submission to publication but we think revision will add to the impact and longevity of your important work.

We look forward to receiving your revised manuscript.

Nāku iti noa, nā,

Dylan A Mordaunt

Academic Editor

PLOS ONE

Journal Requirements:

2. Please provide additional details regarding participant consent. In the Methods section, please ensure that you have specified (1) whether consent was informed and (2) what type you obtained (for instance, written or verbal). If your study included minors, state whether you obtained consent from parents or guardians. If the need for consent was waived by the ethics committee, please include this information.

4. Thank you for stating the following in the Financial Disclosure section: 

"BF, HC, NP, and EC were all funded by a New Zealand Ministry of Business, Innovation, and Employment Endeavour Grant. https://www.mbie.govt.nz/science-and-technology/science-and-innovation/funding-information-and-opportunities/investment-funds/endeavour-fund/success-stories/past-rounds/2016-successful-proposals/ The Funder had no role in study design, data collection and analysis, decision to publish, or preparation of the manuscript." 

We note that one or more of the authors have an affiliation to the commercial funders of this research study : New Zealand Ministry of Business, Innovation, and Employment Endeavour Grant 

5. Please amend your authorship list in your manuscript file to include author Brodie Fraser, Hera Cook, Elinor Chisholm and Nevil Pierse.

Additional Editor Comments:

Thank you for your submission. This is interesting and important work. We are fortunate to have received comprehensive reviews from four reviewers, who have covered a broad range of areas. I think reflecting on these will add value and ultimately increase the impactfulness of the paper. None of these are required per se, but consideration of epistemology, eleborating on the methods and expansion of the discussion would add value. One of the reviewers has commented on the overall need for more detail and the authors are not limited by length in PLoS. It would also be valuable to consider reflection on how the results would impact program and policy development in Aotearoa. The decision has been marked as major revision based on advice from the reviewers and the extent of modifications suggested. However from my perspective as academic editor, I do not have concerns about the manuscript fundamentals and intend to progress this upon receipt of a revision based on the importance and novelty of the work. Ngā mihi.

Reviewers' comments:

Reviewer's Responses to Questions

**Comments to the Author**

1. Is the manuscript technically sound, and do the data support the conclusions?

Reviewer #1: Partly

Reviewer #2: Partly

Reviewer #3: Yes

Reviewer #4: Yes

2. Has the statistical analysis been performed appropriately and rigorously? 

Reviewer #1: N/A

Reviewer #2: N/A

Reviewer #3: N/A

Reviewer #4: N/A

3. Have the authors made all data underlying the findings in their manuscript fully available?

Reviewer #1: No

Reviewer #2: No

Reviewer #3: No

Reviewer #4: Yes

4. Is the manuscript presented in an intelligible fashion and written in standard English?

Reviewer #1: Yes

Reviewer #2: Yes

Reviewer #3: Yes

Reviewer #4: Yes

5. Review Comments to the Author

Reviewer #1: This is an interesting paper that has great potential to make an original contribution to a growing field. The empirical material is clearly rich and allows us to hear the voice of people often not heard in research. think the paper will be benefit from restructuring to be clearer on the contributions of the paper to existing scholarship and what is specifically new here. I would also recommend a reworking of the presentation of the data and more detail and explanation of methods.

Structure: at present the discussion reads more like the outlines of a literature review. It mostly presents what has been established in earlier research and how the current study aligns with and supports this. The particular contributions of this paper could be more clearly outlined if at least some of this material was presented earlier, with the addition of material on gaps in the literature that are relevant to this study: the New Zealand housing policy and market context, for example. The discussion could then focus on illuminating where and how this study contributes new empirical and/or conceptual insights

It would be good to know more about the conceptual approach framing this study and the research questions driving it. What did the study set out to find and why? How does this slice of the bigger study contribute to the larger study and the scholarship more broadly? For example, the article opens by noting the emphasis in the existing literature on LGBTQI+ young people and the importance of attention to adults, but most of the data presented discusses the childhood and adolescent experiences of the (now adult) participants in the study. This isn’t an intractable problem: if the attention of the paper is on the experiences that adults had when they were younger that’s fine, but the framing of the paper then needs to be something other than it is. If, on the other hand, the attention of the paper is on young people and adults, as claimed, that’s fine too: but we need clearer arguments about the data that relates to young people and that which relates to adults, and greater conceptual detail on the themes as they relate to the two groups.

Methods

More information is needed on:

• Recruitment: please say more about ‘additional key locations’ and how you decided which were key locations to use and which to avoid? How did you recruit people who were visible as homeless because attending key locations, but not in contact with housing service providers?

• Interviews: what were the interviews about, what were the inclusion criteria, how was screening done, how long did interviews take, how was it determined that the information for which probing could be done was important enough to warrant another interview?

• Data analysis: the section called ‘data analysis’ is about sampling and sample size, not data analysis. Much more detail needed on data analysis, including citations: what was the rationale for thematic analysis , and which type(s) of thematic analysis was used? How were themes derive from the data. And on the question of sampling: more information is needed on this. How did you know that theoretical sufficiency had been reached after 7 seven interviews?

• Sample: does Table 1 indicate that, for example, one participant (Thom) had experienced homelessness in the 1990s and not since? Could this be specified if so, and could the age(s) of participants when experiencing homelessness rather than the decade be used? Also Table 1: how was the social class of participants determined? Who determined it?

Presentation of data

‘Instability’ is a central concept to the findings, and more detail on how this was derived, context in the scholarship, and perspectives of participants would be helpful. Overall, it’s not clear when/if instability as a theme was generated from data, or the participants' own constructions.

More generally, most of the quotes presented are exegesis of circumstances and events, rather than data that would illustrate or illuminate the themes. It would be more interesting and robust to paraphrase these circumstances rather and use verbatim language from the interviews. As one example (but this happens throughout), the description of Felix’s experiences on p10 (feeling as though he lacked autonomy; that it took a long time to him to feel as though he had control about how his life went etc) would be stronger if it was supported by data, the account of him starting work and living independently does not need a quote in the same way. Similarly on p9, Mariella ‘felt she was not able to be a child’—the quote doesn’t really support this as it doesn’t say anything about Mariella’s feelings about her childhood. More evidence would be helpful.

More specific points: I’d like to see more definitional discussion about what adutltfication is, as some of the discussion here seems to me about adverse experiences that aren’t generally referred to as adultification. E.g., the discussion of Nico’s agency around sex and consent on p11 is powerful but I don’t see how adultification quite fits. My understanding is that the idea is useful for describing not early exposure to events as such, but a process whereby roles and relationships are imposed and taken on.

The discussion on housing experiences on p13 is interesting and resonates with other studies on tight housing markets and the difficulties people face—I think more of a link is needed between the accounts given here and homelessness. Lowered expectations, insecure and inadequate housing, and conflict with landlords are challenging and often unjust experiences, but the connections between these participants’ experiences and their later homelessness are not yet explicit enough. Similarly, the discussion on pp14-15 about stigma and hiding sexual identity is important (although again not really new) but the connection with homelessness isn’t apparent.

Readers unfamiliar with tenancy laws in New Zealand won’t know if the eviction process Clara was describing is legal (i.e., are no fault evictions easy for landlords to implement). I take your point that the situation could have been more complicated than the necessarily subjective account given by Clara, but could the landlord have evicted her as described if they wanted to?

I like very much the argument and approach of systems failures (and would suggest that intersectionality theories and studies could also be useful here) but the data presented doesn’t really, for me, illuminate or show this. If there are examples or pathways that show this it would benefit the argument to present it more directly.

I hope these comments are helpful, and congratulations on the research.

Reviewer #2: Thank you for the opportunity to review this manuscript. It focuses on an important topic that has received little attention in the broader scholarship on homelessness. The authors should be commended for conducting this important work. The manuscript is good, particularly the Results section. It can be enhanced in a few important ways, particularly through a more detailed descriptions of the data analysis and through discussing the implications of this research. Below are some specific points for consideration:

Introduction

I was unaware of the word “Takatāpui”. Thank you for sharing this information with me!

The definitions of LGBTIQ+ and Takatāpui could be included as footnotes, rather than directly in the text. It disrupts the flow as presented.

Page 2 – What do the authors mean by “ethnicity and racism”? Racism based upon one’s ethnicity/cultural identity?

In the NZ Census, does the data only include cisgender women and men? Or are there statistics on transgender/non-binary/gender non-conforming individuals?

Methods

Although the explanation as to why soup kitchens and night shelters were not visited is valid, it does restrict the representativeness of the sample.

The first author should also recognize that their own biases may have impacted the selection criteria for the recruitment locations.

What is the NZ definition of homelessness?

Were there people interested in participating who later declined? Or were there only eight people who agreed to participate? It would be interesting to know the potential sample size and the actual sample size.

Were participants compensated for their participation?

Were the interviews recorded? Were the interviews transcribed?

Data Analysis

This section needs to provide much greater detail on the analysis.

Who conducted the data analysis? How was the data coded? How were themes developed? How was the data reviewed across the transcripts? What steps did the authors take to establish the trustworthiness of the data and analysis? How was subjectivity addressed in the analysis? Was there any form of member checking?

Results

This section is very well done. It is able to present the stories of participants and identifies common threads in a very clear and coherent manner.

It would be helpful to define what is meant by “middle class” and “upper-middle class”. Were these defined by the researchers or the participants?

Page 6 – Are these categories or themes?

The “Difficulty in Find Housing” section is good, but it appears that many of the stories presented are not necessarily about prior experiences to becoming homeless, but the continued difficulties to find appropriate and affordable housing. Did participants’ negative experiences with landlords and the lack of affordable housing result into them experiencing homelessness? The linkage is not always clear in this section.

With this section, what makes the experiences of LGBTIQ+ and Takatāpui people different from non- LGBTIQ+ and Takatāpui people experiencing homelessness?

Discussion and Conclusion

These sections are good, but both could benefit from discussing the implications of these findings. What programs and policies need to be developed to prevent LGBTIQ+ and Takatāpui homelessness and support LGBTIQ+ and Takatāpui people who experience homelessness?

It could be helpful to review the “Pathways into Homelessness Among LGBTQ+ Adults” paper by Ecker, Aubry, and Sylvestre (2020).

What are the limitations of the study?

What future research should happen?

Table 1 is presented twice.

Reviewer #3: Thank you for the opportunity to review this paper. This is, to the best of my knowledge, an original study in the New Zealand context and I am confident that the study has international relevance. The authors provide powerful material, drawn from a larger study, that provides insights into the experiences of takatāpui/LGBTIQ+ people before becoming homeless. The study’s aims, methods, results and conclusions are all presented clearly.

The paper is very long (no word count is provided). I note that the results section contains discussion of the results including reference to the relevant literature; from a reader perspective this works, but the editors may have a preference as to how qualitative research findings and the discussion are presented (together or separately).

The methodology appears sound. I have only one relatively minor query. In the methods section, please explain how socioeconomic background was classified. For example, what does “upper middle class” mean, as distinct from “middle class”?

The discussion section is relatively brief and lacks a description of the study’s methodological strengths and weaknesses. This omission needs to be fixed. Similarly there is no mention of future research priorities. The authors might like to give this latter point consideration. The conclusions section of the paper does not offer policy recommendations; probably these are not warranted in this paper, but the authors may wish to signal the importance of the issues raised in this paper from a policy perspective in somewhat more detail.

Minor changes

Page 2: In the abstract, provide in parenthesis an example of what “pervasiveness of instability” means to assist comprehension.

Page 2, abstract: replace the ; after “categories of” with :

Page 2, introduction line 1: delete the s at the end of “remains”

Page 2, introduction line 2: delete “more broadly”

Page 2, introduction line 11: delet “enough”

Page 3, paragraph 2, line 5: replace the ; with :

Page 3, paragraph 2, line 9: replace the ; with :

Page 4, line 1: delete “and”

Page 4: “…that there is limited institutional trust with organisations meant to provide support.” Please replace “with” with “of” to make this sentence clearer. Also, is the word “institutional” necessary, or does it confuse the sentence?

Pages 3 and 5: The footnote related to the meaning of “Pākehā” is replicated.

Page 8, line 5: replace the ; with ,

Page 8, line 11: “lead” should be “led”

Page 10 line 6: replace the ; with :

Page 16: replace the ; with ,

Page 20, third paragraph: “…becoming homelessness” should be “…becoming homeless”

Page 20, discussion first paragraph: is “elevates” the right word? Would “increases” be better?

Reviewer #4: Review PONE-D-21-26555

The manuscript presents a very emerging theme, using data from semi-structured interviews with eight homeless people in Aotearoa New Zealand (NZ) and belonging to Takatāpui / LGBTIQ +.

Based on their respective previous experiences, the authors managed to create a line of reasoning that brings together several key elements that contribute to understanding the reasons why housing instability is very frequent in the LGBTI population.

It is possible to identify several previous instability factors experienced by these people, which act in an intersectional way, leading as a first consequence to the need to leave childhood and adolescence early, without having psychic resources to understand the events they are experiencing. Thus, the impact of instability is lasting and propagates throughout the person's life, ultimately leading to structural 'disempowerment' and marginality.

The authors justify that even though the last demographic census carried out in NZ is recent (2018), for the Takatāpui/LGBTIQ+ population, statistics on housing instability situation do not exist.

It also adds a discussion on norms and legislation in the area of social assistance that also contribute to these findings, including characteristics of the local real estate system, in addition to the feeling of being discriminated against or even suffering judgments from the homeowners, with reports of open discrimination.

The results are well described, and in summary, it is clear why the difficulty in finding housing was revealed as a key category throughout the interviews. It is very interesting to observe the strategies operated by the interviewees to try to overcome this difficulty, which includes omitting their own gender identity.

To further qualify the article, I suggest that references on intersectionality be included in the introduction and discussion, in order to consolidate the interpretation of the findings and allow for the organization of recommendations to overcome this complex problem and consolidate public policies in favor of this population.

6. PLOS authors have the option to publish the peer review history of their article (what does this mean?). If published, this will include your full peer review and any attached files.

Reviewer #1: No

Reviewer #2: No

Reviewer #3: **Yes: **Peter Crampton

Reviewer #4: **Yes: **Katia Cristina Bassichetto

---

## [Author Response · Author response to Decision Letter 0]

5 Oct 2021

“You’re So Powerless”: Takatāpui/LGBTIQ+ People’s Experiences Before Becoming Homeless in Aotearoa New Zealand – Response to reviewers

Tēnā koe,

We thank the reviewers for their thoughtful feedback on our paper, and the editor for his comments and work finding reviewers for it. We have copied reviewer comments into this document and responded in bullet points underneath each separate comment. We have also addressed the editor’s comments (including the issue with the funding statement) below.

We note that the paper has been lengthened considerably with the required changes made (although a number were the addition of new references!). We know that the Journal does not have a word limit for papers, but are open to shortening the manuscript if the reviewers and/or editor deem it necessary. We feel that this length has allowed us to create valuable changes to the paper, congruent with the reviewers’ comments.

We would like to note that in the document with tracked changes there is currently an issue with the footnote numbering whereby there is no footnote numbered 2. This is due to the tracked changes and moving of Takatāpui/LGBTIQ+ definitions to a footnote (it was where the 2nd footnote was previously located). When this change is accepted the numbering of the footnotes are fixed.

Ngā mihi nui,

The authors. 

Editor Feedback

Journal Requirements:

• Apologies for this mistake, we have amended the documents to suit.

2. Please provide additional details regarding participant consent. In the Methods section, please ensure that you have specified (1) whether consent was informed and (2) what type you obtained (for instance, written or verbal). If your study included minors, state whether you obtained consent from parents or guardians. If the need for consent was waived by the ethics committee, please include this information.

• We have added a sentence about informed consent and the type we obtained. We note there were no minors who participated in the study.

• We cannot find the “funding information” section in the submission, only the “financial disclosure” section. Would the editor be able to assist us with this?

4. Thank you for stating the following in the Financial Disclosure section: 

"BF, HC, NP, and EC were all funded by a New Zealand Ministry of Business, Innovation, and Employment Endeavour Grant. https://www.mbie.govt.nz/science-and-technology/science-and-innovation/funding-information-and-opportunities/investment-funds/endeavour-fund/success-stories/past-rounds/2016-successful-proposals/ The Funder had no role in study design, data collection and analysis, decision to publish, or preparation of the manuscript." 

We note that one or more of the authors have an affiliation to the commercial funders of this research study : New Zealand Ministry of Business, Innovation, and Employment Endeavour Grant 

• Apologies for the confusion with this. MBIE do not directly pay salaries, and none of us have commercial affiliations with them. They are one of the main research funders in Aotearoa New Zealand, who are responsible for government funding of science. Would it be sufficient to amend the financial disclosure to read “This research was funded by a grant from the New Zealand Ministry of Business, Innovation, and Employment Endeavour Fund. https://www.mbie.govt.nz/science-and-technology/science-and-innovation/funding-information-and-opportunities/investment-funds/endeavour-fund/success-stories/past-rounds/2016-successful-proposals/ The Funder had no role in study design, data collection and analysis, decision to publish, or preparation of the manuscript. The specific roles of these authors are articulated in the ‘author contributions’ section. None of the authors were paid directly by the funder; funding was paid to the authors’ University, who paid BF’s PhD scholarship and HC, EC, and NP’s salaries.”? Or we could leave out the final sentence about the University if the rest of the statement is sufficient. Our University also had no role in in the study design, data collection and analysis, decision to publish, or preparation of the manuscript.

• As noted above, we have no commercial affiliation with the funder. 

5. Please amend your authorship list in your manuscript file to include author Brodie Fraser, Hera Cook, Elinor Chisholm and Nevil Pierse.

• We have added our separate title page document to the manuscript.

Additional Editor Comments:

Thank you for your submission. This is interesting and important work. We are fortunate to have received comprehensive reviews from four reviewers, who have covered a broad range of areas. I think reflecting on these will add value and ultimately increase the impactfulness of the paper. None of these are required per se, but consideration of epistemology, eleborating on the methods and expansion of the discussion would add value. One of the reviewers has commented on the overall need for more detail and the authors are not limited by length in PLoS. It would also be valuable to consider reflection on how the results would impact program and policy development in Aotearoa. The decision has been marked as major revision based on advice from the reviewers and the extent of modifications suggested. However from my perspective as academic editor, I do not have concerns about the manuscript fundamentals and intend to progress this upon receipt of a revision based on the importance and novelty of the work. Ngā mihi.

• Thank you for your comments. As you will see below we have happily taken on board the reviewers comments and are appreciative of the time everyone has spent on our manuscript. We agree that these have added value to the paper.

Reviewer #1 

This is an interesting paper that has great potential to make an original contribution to a growing field. The empirical material is clearly rich and allows us to hear the voice of people often not heard in research. think the paper will be benefit from restructuring to be clearer on the contributions of the paper to existing scholarship and what is specifically new here. I would also recommend a reworking of the presentation of the data and more detail and explanation of methods. Structure: at present the discussion reads more like the outlines of a literature review. It mostly presents what has been established in earlier research and how the current study aligns with and supports this. The particular contributions of this paper could be more clearly outlined if at least some of this material was presented earlier, with the addition of material on gaps in the literature that are relevant to this study: the New Zealand housing policy and market context, for example. The discussion could then focus on illuminating where and how this study contributes new empirical and/or conceptual insights

• We have added a paragraph to the discussion on the context of the welfare state in NZ, and provided policy recommendations at the end of this paragraph too. We have highlighted the need for the NZ welfare state to better cater to Takatāpui/LGBTIQ+ people.

It would be good to know more about the conceptual approach framing this study and the research questions driving it. What did the study set out to find and why? How does this slice of the bigger study contribute to the larger study and the scholarship more broadly? For example, the article opens by noting the emphasis in the existing literature on LGBTQI+ young people and the importance of attention to adults, but most of the data presented discusses the childhood and adolescent experiences of the (now adult) participants in the study. This isn’t an intractable problem: if the attention of the paper is on the experiences that adults had when they were younger that’s fine, but the framing of the paper then needs to be something other than it is. If, on the other hand, the attention of the paper is on young people and adults, as claimed, that’s fine too: but we need clearer arguments about the data that relates to young people and that which relates to adults, and greater conceptual detail on the themes as they relate to the two groups.

• We have included the research aims in the introduction for clarity.

• We have ensured that the opening sentences for each section of the results notes whether the experiences discussed in that section relate to youth and/or adulthood. 

Methods

More information is needed on:

Recruitment: please say more about ‘additional key locations’ and how you decided which were key locations to use and which to avoid? How did you recruit people who were visible as homeless because attending key locations, but not in contact with housing service providers?

• We have added the names of these key locations. They were not housing organisations, but organisations which provided support and community connections.

Interviews: what were the interviews about, what were the inclusion criteria, how was screening done, how long did interviews take, how was it determined that the information for which probing could be done was important enough to warrant another interview?

• We have added this further detail in a new paragraph in the methods section, and several new footnotes in the same section.

Data analysis: the section called ‘data analysis’ is about sampling and sample size, not data analysis. Much more detail needed on data analysis, including citations: what was the rationale for thematic analysis , and which type(s) of thematic analysis was used? How were themes derive from the data. And on the question of sampling: more information is needed on this. How did you know that theoretical sufficiency had been reached after 7 seven interviews?

• We have added greater detail to the data analysis section, including information on how the transcripts were coded. Additionally, we have included references about the use of constructivist grounded theory.

Sample: does Table 1 indicate that, for example, one participant (Thom) had experienced homelessness in the 1990s and not since? Could this be specified if so, and could the age(s) of participants when experiencing homelessness rather than the decade be used? Also Table 1: how was the social class of participants determined? Who determined it?

• We have purposefully decided to only present the decade/s in which participants experienced homelessness as we are concerned that if we include their current age, with the exact age at which they experienced homelessness, that their anonymity could be jeopardised. Aotearoa New Zealand is a small country, and our Takatāpui/LGBTIQ+ community is even smaller.

• We have added a footnote (#12) explaining how class is used and determined in this research.

‘Instability’ is a central concept to the findings, and more detail on how this was derived, context in the scholarship, and perspectives of participants would be helpful. Overall, it’s not clear when/if instability as a theme was generated from data, or the participants' own constructions.

• We have clarified that this was generated both from data and the participants’ own construction. We have also added a sentence in the introduction to this section outlining connections to the literature.

More generally, most of the quotes presented are exegesis of circumstances and events, rather than data that would illustrate or illuminate the themes. It would be more interesting and robust to paraphrase these circumstances rather and use verbatim language from the interviews. As one example (but this happens throughout), the description of Felix’s experiences on p10 (feeling as though he lacked autonomy; that it took a long time to him to feel as though he had control about how his life went etc) would be stronger if it was supported by data, the account of him starting work and living independently does not need a quote in the same way. Similarly on p9, Mariella ‘felt she was not able to be a child’—the quote doesn’t really support this as it doesn’t say anything about Mariella’s feelings about her childhood. More evidence would be helpful. More specific points: I’d like to see more definitional discussion about what adultification is, as some of the discussion here seems to me about adverse experiences that aren’t generally referred to as adultification. E.g., the discussion of Nico’s agency around sex and consent on p11 is powerful but I don’t see how adultification quite fits. My understanding is that the idea is useful for describing not early exposure to events as such, but a process whereby roles and relationships are imposed and taken on.

• We have added information further defining adultification in the introduction to this section. We have also expanded on the examples given and more directly linked them to adultification. However, we have kept the direct quotes as we believe this style is more in keeping with the rest of the paper. We believe the additions we have made have resolved these issues the reviewer highlighted, but we are happy to make further amendments if the reviewer does not agree. 

The discussion on housing experiences on p13 is interesting and resonates with other studies on tight housing markets and the difficulties people face—I think more of a link is needed between the accounts given here and homelessness. Lowered expectations, insecure and inadequate housing, and conflict with landlords are challenging and often unjust experiences, but the connections between these participants’ experiences and their later homelessness are not yet explicit enough. 

• We have made amendments throughout this section to highlight these connections in a clearer manner. 

Similarly, the discussion on pp14-15 about stigma and hiding sexual identity is important (although again not really new) but the connection with homelessness isn’t apparent. Readers unfamiliar with tenancy laws in New Zealand won’t know if the eviction process Clara was describing is legal (i.e., are no fault evictions easy for landlords to implement). I take your point that the situation could have been more complicated than the necessarily subjective account given by Clara, but could the landlord have evicted her as described if they wanted to?

• We have added a few sentences throughout this section to make clearer the links with homelessness. Unfortunately it is hard for us to comment on the legal setting regarding evictions as the interview was unclear as to exactly what period of time Clara was referring to: we believe it could have been anywhere between the 1990’s and 2010’s. It is highly likely the landlord could have undertaken a no-cause eviction; we have added a footnote about this (#20).

I like very much the argument and approach of systems failures (and would suggest that intersectionality theories and studies could also be useful here) but the data presented doesn’t really, for me, illuminate or show this. If there are examples or pathways that show this it would benefit the argument to present it more directly.

• We have expanded paragraphs throughout this section to better show how the examples relate to systems failures. We have also added additional references to intersectionality (and as per reviewer 4’s request we have added references to intersectionality theories throughout the paper).

I hope these comments are helpful, and congratulations on the research.

Reviewer #2

Thank you for the opportunity to review this manuscript. It focuses on an important topic that has received little attention in the broader scholarship on homelessness. The authors should be commended for conducting this important work. The manuscript is good, particularly the Results section. It can be enhanced in a few important ways, particularly through a more detailed descriptions of the data analysis and through discussing the implications of this research. Below are some specific points for consideration:

Introduction

I was unaware of the word “Takatāpui”. Thank you for sharing this information with me!

The definitions of LGBTIQ+ and Takatāpui could be included as footnotes, rather than directly in the text. It disrupts the flow as presented.

• We have changed this to a footnote (#1).

Page 2 – What do the authors mean by “ethnicity and racism”? Racism based upon one’s ethnicity/cultural identity?

• We have amended the wording of this to “racism and ethnic discrimination” for clarity.

In the NZ Census, does the data only include cisgender women and men? Or are there statistics on transgender/non-binary/gender non-conforming individuals?

• The reviewer is correct in presuming the NZ Census doesn’t identify people who are transgender, gender diverse, and gender non-conforming. We have added a footnote about this (#3).

Methods

Although the explanation as to why soup kitchens and night shelters were not visited is valid, it does restrict the representativeness of the sample.

The first author should also recognize that their own biases may have impacted the selection criteria for the recruitment locations.

• We added to this section, including noting the use of the critical paradigm to guide this research and how it provides space for the researcher’s values.

What is the NZ definition of homelessness?

• We have added a sentence explaining the NZ definition of homelessness in the introduction paragraph outlining the NZ context. 

Were there people interested in participating who later declined? Or were there only eight people who agreed to participate? It would be interesting to know the potential sample size and the actual sample size.

• We have included these additional details in a new paragraph (and edits to existing paragraphs) in the methods section.

Were participants compensated for their participation?

• We have included these additional details in a new paragraph (and edits to existing paragraphs) in the methods section.

Were the interviews recorded? Were the interviews transcribed? 

• We have included these additional details in a new paragraph (and edits to existing paragraphs) in the methods section.

Data Analysis

This section needs to provide much greater detail on the analysis. Who conducted the data analysis? How was the data coded? How were themes developed? How was the data reviewed across the transcripts? What steps did the authors take to establish the trustworthiness of the data and analysis? How was subjectivity addressed in the analysis? Was there any form of member checking?

• As mentioned in response to reviewer 1, we have added greater detail to the data analysis section, including information on how the transcripts were coded and the contributions each author made during this process.

Results

This section is very well done. It is able to present the stories of participants and identifies common threads in a very clear and coherent manner.

It would be helpful to define what is meant by “middle class” and “upper-middle class”. Were these defined by the researchers or the participants?

• As noted in response to other reviewer’s comments, we have added a footnote (#12) explaining how class is used and determined in this research.

Page 6 – Are these categories or themes?

• These are categories as per grounded theory conventions; we used grounded theory to analyse the data, not thematic analysis. We have expanded on how we used grounded theory in the amended methods section.

The “Difficulty in Find Housing” section is good, but it appears that many of the stories presented are not necessarily about prior experiences to becoming homeless, but the continued difficulties to find appropriate and affordable housing. Did participants’ negative experiences with landlords and the lack of affordable housing result into them experiencing homelessness? The linkage is not always clear in this section. With this section, what makes the experiences of LGBTIQ+ and Takatāpui people different from non- LGBTIQ+ and Takatāpui people experiencing homelessness?

• We have made edits throughout this section to further clarify the links.

Discussion and Conclusion

These sections are good, but both could benefit from discussing the implications of these findings. What programs and policies need to be developed to prevent LGBTIQ+ and Takatāpui homelessness and support LGBTIQ+ and Takatāpui people who experience homelessness?

• As will be discussed below in response to reviewer 3, we have added policy suggestions where appropriate. We note that we do not include policy recommendations that apply to experiences specifically during and after homelessness; we hope to include such recommendations in future papers we publish about the participants’ experiences during and after homelessness. 

It could be helpful to review the “Pathways into Homelessness Among LGBTQ+ Adults” paper by Ecker, Aubry, and Sylvestre (2020).

• We really enjoy Ecker and colleague’s work in this space. We have added a sentence referencing this paper in the first paragraph of the discussion. We have kept mention of it to that as our paper discusses experiences in both childhood and adulthood, and does not explicitly focus on direct pathways into homelessness; instead we look at lifetime experiences prior to becoming homeless which contribute to these experiences (such as having to grow up fast).

What are the limitations of the study?

• We have added a paragraph into the discussion that covers strengths, limitations, and avenues for future research.

What future research should happen?

• We have added a paragraph into the discussion that covers strengths, limitations, and avenues for future research.

Table 1 is presented twice.

• We believe this is due to the different required documents for submission – we included table 1 in our manuscript and a separate file with just the table in it as per journal requirements. Table one only appears in the manuscript itself once.

Reviewer #3

Thank you for the opportunity to review this paper. This is, to the best of my knowledge, an original study in the New Zealand context and I am confident that the study has international relevance. The authors provide powerful material, drawn from a larger study, that provides insights into the experiences of takatāpui/LGBTIQ+ people before becoming homeless. The study’s aims, methods, results and conclusions are all presented clearly.

The paper is very long (no word count is provided). I note that the results section contains discussion of the results including reference to the relevant literature; from a reader perspective this works, but the editors may have a preference as to how qualitative research findings and the discussion are presented (together or separately).

• We note that the paper is indeed long (over 13,000 words); we were conscious though when submitting it that the first draft be under 10,000 words (which it was, and we apologise for neglecting to include that detail in our manuscript) in order to be consistent with what a number of other qualitative journals allow. 

• We agree that the discussion of results could be moved entirely to the discussion, but feel that now we have amended the discussion as asked for by other reviewers, that the article flows well with the existing structure. It is also easier for the reader to follow some of our points being made when we delve into discussing the results alongside the quotes we have presented. 

The methodology appears sound. I have only one relatively minor query. In the methods section, please explain how socioeconomic background was classified. For example, what does “upper middle class” mean, as distinct from “middle class”?

• As noted in response to other reviewer’s comments, we have added a footnote (#5) explaining how class was used and determined in this research.

The discussion section is relatively brief and lacks a description of the study’s methodological strengths and weaknesses. This omission needs to be fixed. Similarly there is no mention of future research priorities. The authors might like to give this latter point consideration. The conclusions section of the paper does not offer policy recommendations; probably these are not warranted in this paper, but the authors may wish to signal the importance of the issues raised in this paper from a policy perspective in somewhat more detail.

• As mentioned in response to reviewer 2, we have added a paragraph into the discussion that covers strengths, limitations, and avenues for future research. We have also added in some policy recommendations where relevant (although noting that there are many other policy recommendations we have which better relate to future papers we hope to publish looking at experiences during and after homelessness amongst this group).

Minor changes

Page 2: In the abstract, provide in parenthesis an example of what “pervasiveness of instability” means to assist comprehension.

Page 2, abstract: replace the ; after “categories of” with :

Page 2, introduction line 1: delete the s at the end of “remains”

Page 2, introduction line 2: delete “more broadly”

Page 2, introduction line 11: delet “enough”

Page 3, paragraph 2, line 5: replace the ; with :

Page 3, paragraph 2, line 9: replace the ; with :

Page 4, line 1: delete “and”

Page 4: “…that there is limited institutional trust with organisations meant to provide support.” Please replace “with” with “of” to make this sentence clearer. Also, is the word “institutional” necessary, or does it confuse the sentence?

Pages 3 and 5: The footnote related to the meaning of “Pākehā” is replicated.

Page 8, line 5: replace the ; with ,

Page 8, line 11: “lead” should be “led”

Page 10 line 6: replace the ; with :

Page 16: replace the ; with ,

Page 20, third paragraph: “…becoming homelessness” should be “…becoming homeless”

Page 20, discussion first paragraph: is “elevates” the right word? Would “increases” be better?

• Thank you for taking the time to note these minor mistakes. We have amended all of them.

Reviewer #4

The manuscript presents a very emerging theme, using data from semi-structured interviews with eight homeless people in Aotearoa New Zealand (NZ) and belonging to Takatāpui / LGBTIQ +.

Based on their respective previous experiences, the authors managed to create a line of reasoning that brings together several key elements that contribute to understanding the reasons why housing instability is very frequent in the LGBTI population. It is possible to identify several previous instability factors experienced by these people, which act in an intersectional way, leading as a first consequence to the need to leave childhood and adolescence early, without having psychic resources to understand the events they are experiencing. Thus, the impact of instability is lasting and propagates throughout the person's life, ultimately leading to structural 'disempowerment' and marginality. The authors justify that even though the last demographic census carried out in NZ is recent (2018), for the Takatāpui/LGBTIQ+ population, statistics on housing instability situation do not exist. It also adds a discussion on norms and legislation in the area of social assistance that also contribute to these findings, including characteristics of the local real estate system, in addition to the feeling of being discriminated against or even suffering judgments from the homeowners, with reports of open discrimination. The results are well described, and in summary, it is clear why the difficulty in finding housing was revealed as a key category throughout the interviews. It is very interesting to observe the strategies operated by the interviewees to try to overcome this difficulty, which includes omitting their own gender identity.

• We thank the reviewer for their comments.

To further qualify the article, I suggest that references on intersectionality be included in the introduction and discussion, in order to consolidate the interpretation of the findings and allow for the organization of recommendations to overcome this complex problem and consolidate public policies in favor of this population.

• We have added several sentences on intersectionality in the introduction and throughout the discussion.

---

## [Editor Report · Decision Letter 1]

27 Oct 2021

“You’re so powerless”: Takatāpui/LGBTIQ+ people’s experiences before becoming homeless in Aotearoa New Zealand

PONE-D-21-26555R1

Tenā koe Dr. Fraser,

We’re pleased to inform you that your manuscript has been judged scientifically suitable for publication and will be formally accepted for publication once it meets all outstanding technical requirements.

Ngā mihi maioha,

Dylan A Mordaunt, MB ChB, FRACP, FAIDH

Academic Editor

PLOS ONE

Additional Editor Comments (optional):

Thank you for your resubmission. I believe the changes more than adequately address the suggestions made and further review is not required.
---

## [Editor Report · Acceptance letter]

9 Dec 2021

PONE-D-21-26555R1 

“You’re so powerless”: Takatāpui/LGBTIQ+ people’s experiences before becoming homeless in Aotearoa New Zealand 

Dear Dr. Fraser:

I'm pleased to inform you that your manuscript has been deemed suitable for publication in PLOS ONE. Congratulations! Your manuscript is now with our production department. 

Kind regards, 

on behalf of

Dr. Dylan A Mordaunt 

Academic Editor

PLOS ONE